# Solving the Boundary Value Problems for Differential Equations with Fractional Derivatives by the Method of Separation of Variables

**Temirkhan Aleroev** 

National Research Moscow State University of Civil Engineering (NRU MGSU), Yaroslavskoe Shosse, 26, 129337 Moscow, Russia; AleroevTS@mgsu.ru

**Abstract:** This paper is devoted to solving boundary value problems for differential equations with fractional derivatives by the Fourier method. The necessary information is given (in particular, theorems on the completeness of the eigenfunctions and associated functions, multiplicity of eigenvalues, and questions of the localization of root functions and eigenvalues are discussed) from the spectral theory of non-self-adjoint operators generated by differential equations with fractional derivatives and boundary conditions of the Sturm–Liouville type, obtained by the author during implementation of the method of separation of variables (Fourier). Solutions of boundary value problems for a fractional diffusion equation and wave equation with a fractional derivative are presented with respect to a spatial variable.

**Keywords:** eigenvalue; eigenfunction; function of Mittag–Leffler; fractional derivative; Fourier method; method of separation of variables

---

*In memoriam of my Father Sultan*
*and my son Bibulat*

## 1. Introduction

Let $\varphi(x) \in L_1(0,1)$. Then the function

$$\frac{d^{-\alpha}}{dx^{-\alpha}}\varphi(x) \equiv \frac{1}{\Gamma(\alpha)}\int_0^x (x-t)^{\alpha-1}\varphi(t)\,dt \in L_1(0,1)$$

is known as a fractional integral of order $\alpha > 0$ beginning at $x = 0$ [1]. Here $\Gamma(\alpha)$ is the Euler gamma-function. As is known (see [1]), the function $\psi(x) \in L_1(0,1)$ is called the fractional derivative of the function $\varphi(x) \in L_1(0,1)$ of order $\alpha > 0$ beginning at $x = 0$, if

$$\varphi(x) = \frac{d^{-\alpha}}{dx^{-\alpha}}\psi(x),$$

which is written

$$\psi(x) = \frac{d^{\alpha}}{dx^{\alpha}}\varphi(x).$$

Then

$$\frac{d^{\alpha}}{dx^{\alpha}}$$

denote the fractional integral for $\alpha < 0$ and the fractional derivative for $\alpha > 0$.

Let $\{\gamma_k\}_0^n$ be a set of real numbers satisfying the condition $0 < \gamma_j \leq 1$, $(0 \leq j \leq n)$. We denote

$$\sigma_k = \sum_{j=0}^{k} \gamma_j - 1;$$

$$\mu_k = \sigma_k + 1 = \sum_{j=0}^{k} \gamma_j \ (0 \leq k \leq n),$$

and assume that

$$\frac{1}{\rho} = \sum_{j=0}^{n} \gamma_j - 1 = \sigma_n = \mu_n - 1 > 0.$$

Following M. M. Dzhrbashyan [1], we consider the integro-differential operators

$$D^{(\sigma_0)}\varphi(x) \equiv \frac{d^{-(1-\gamma_0)}}{dx^{-(1-\gamma_0)}}\varphi(x),$$

$$D^{(\sigma_1)}\varphi(x) \equiv \frac{d^{-(1-\gamma_1)}}{dx^{-(1-\gamma_1)}}\frac{d^{\gamma_0}}{dx^{\gamma_0}}\varphi(x),$$

$$D^{(\sigma_2)}\varphi(x) \equiv \frac{d^{-(1-\gamma_2)}}{dx^{-(1-\gamma_2)}}\frac{d^{\gamma_1}}{dx^{\gamma_1}}\frac{d^{\gamma_0}}{dx^{\gamma_0}}\varphi(x),$$

$$\cdots\cdots\cdots\cdots$$

$$D^{(\sigma_n)}\varphi(x) \equiv \frac{d^{-(1-\gamma_n)}}{dx^{-(1-\gamma_n)}}\frac{d^{\gamma_{n-1}}}{dx^{\gamma_{n-1}}}\cdots\frac{d^{\gamma_0}}{dx^{\gamma_0}}\varphi(x).$$

We denote $D_{ax}^{\alpha}$ the operator of fractional integro-differentiation of order $\alpha$ beginning at $a \in R$ and with end at $x \in R$ of order $[\alpha]$. By definition we have

$$D_{ax}^{(\alpha)}\phi(t) = \begin{cases} \frac{sign(x-a)}{\Gamma(-\alpha)}\int_a^x \frac{\phi(t)dt}{(x-t)^{\alpha-1}}, & \alpha < 0, \ \phi(t) \in L_1[a,b], \\ \phi(t), \alpha = 0, \ \phi(t) \in L_1[a,b], \\ sign^{[\alpha]+1}(x-a)\frac{\partial^{[\alpha]+1}}{\partial x^{[\alpha]+1}}D_{ax}^{\alpha-[\alpha]-1}\phi(t), & \alpha > 0, \ \phi(t) \in L_1[a,b], \end{cases}$$

where $[\alpha]$ is the integer part of $\alpha$, which satisfies $[\alpha] \leq \alpha < [\alpha] + 1$, and $x \in [a,b]$.

*Boundary Value Problems for Differential Equations of Fractional Order*

The paper is devoted to the method of separation of variables (the Fourier method). This method, which is so widely used in solving boundary value problems for partial differential equations of integer order, until recently remained unsuitable for solving boundary value problems for differential equations with fractional derivatives. The main reason, of course, is that the spectral theory of non-self-adjoint operators generated by the corresponding differential expressions of fractional order and boundary conditions of the Sturm–Liouville type has been supplemented with the necessary information quite recently.

Almost all of the author's papers, to varying degrees, are devoted to the study of the spectral structure of these operators, which constitute the theoretical basis of the method of separation of variables, and the author of this paper, in a sense, summarizes his work in this area.

So, in this paper, a method of separation of variables is presented for solving boundary value problems for differential equations with fractional derivatives of the form

$$\frac{\partial^2 u}{\partial t^2} = \frac{\partial^2 u}{\partial x^2} + C_1 D_{0x}^{\alpha}u + C_0 D_{0t}^{\beta}u, \ 0 < \alpha, \ \beta < 2, \tag{1}$$

and

$$\frac{\partial u(x,t)}{\partial t} = D_{0+}^{\alpha} u(x,t).$$

First of all, we note that anomalous diffusion or dispersion we can describe using the fractional space derivatives, and some processes with 'memory' effects-using the fractional time derivatives.

It should be noted that depending on the modeled process, the fractional differentiation operators appearing in these equations can be both the Riemann–Liouville fractional differentiation operators and the fractional differentiation operators in the Caputo sense. One of the most important problems in modeling physical processes using differential equations with fractional derivatives is the problem of establishing in what sense the fractional derivative is taken and the identification of the order of this fractional derivative.

Undoubtedly, the most significant, fundamental point in the study of boundary value problems for these equations by the method of separation of variables is the question of completeness of systems of eigenfunctions of boundary value problems for the equations

$$L(u; \gamma_0, \gamma_1, \gamma_2, q(x)) = D^{(\sigma_2)} u - [\lambda + q(x)] u(x) = 0, \tag{2}$$

$$X''(x) + C_1 D_{0x}^{\alpha} X = \lambda X(x), \tag{3}$$

(these equations arise when the variables are separated in Equations (2) and (3)).

Therefore, we present basic results from the spectral theory of operators generated by differential equations of the form (2) and boundary conditions of the Sturm–Liouville type.

The relationship between eigenvalues and zeros of a Mittag–Leffler function is shown.

The Green's functions of boundary value problems for equations of the form (2) are considered in detail (it should be noted that these Green's functions were first obtained by the author in his post-graduate student paper [2]), the study of which made it possible to approach problems of the distribution of zeros of a function of the Mittag type from completely new positions-Leffler and reveal the deeply hidden properties of these functions, which for many years have not been possible for specialists in the theory of functions. First of all, we note that the asymptotic properties of the Mittag–Leffler function have been sufficiently well studied [1], but the study of the non-asymptotic properties of the zeros of the Mittag–Leffler function or, similarly, the eigenvalues of operators generated by boundary value problems for Equation (2), is conjugate with large analytical difficulties (in particular, M. M. Dzhrbashian wrote in [1] that "the question about the completeness of the eigenfunctions of boundary value problems for Equation (2) or a finer question about whether these systems compose a basis in $L_2(0,1)$ has a certain interest. However, their solution is apparently associated with significant analytic difficulties".). Therefore, the author gives these properties in sufficient detail.

## 2. Boundary Value Problems for the Fractional Order Diffusion Equation

In this section we present the necessary information from the spectral theory of operators generated by differential equations of fractional order and boundary conditions of the Sturm–Liouville type.

*2.1. Spectral Analysis of Operators, Generated by Fractional Differential Equations of Order More than 1 but Less than 2 and Boundary Conditions of Sturm–Liouville Type and on One Method for Identifying the Order of the Fractional Derivative*

We devote this subsection to the spectral analysis of two boundary value problems [3,4]

$$L(u; 1, 1-\alpha, 1, 0) = \frac{1}{\Gamma(\alpha)} \frac{d}{dx} \int_0^x \frac{u'(\zeta)}{(x-\zeta)^{1-\alpha}} d\zeta + \lambda u(x) = 0$$

$$u(0) = 0, u(1) = 0,$$

and

$$L(u; 1, 1, 1 - \alpha, 0) = \frac{1}{\Gamma(\alpha)} \int_0^x (x - \zeta)^{\alpha-1} u''(\zeta) d\zeta + \lambda u(x) = 0,$$

$$u(0) = 0, u(1) = 0.$$

These problems are the focus of many researchers.

First, we note that in [4] (and references therein), the following problem was considered

$$u'' + \lambda \frac{d^\alpha}{dx^\alpha} u = 0, 0 < \alpha < 1, \tag{4}$$

$$u(0) = 0, u(1) = 0. \tag{5}$$

with studying of the spectrum of the operator

$$D^{(\beta)} u = \frac{d^{-\alpha}}{dx^{-\alpha}} \frac{d^2}{dx^2} u = \frac{1}{\Gamma(\alpha)} \int_0^x (x - \zeta)^{\alpha-1} u''(\zeta) d\zeta, \ (\beta = 2 - \alpha)$$

(the operator $D^{(\sigma_2)}$ transforms to the operator $D^{(\beta)}$ if $\gamma_0 = \gamma_1 = 1$ and $\gamma_2 = 1 - \alpha$).

The operator $D^{(\beta)}$ arose great interest after F. Mainardi's paper [3]. In this paper, the following equation was considered

$$\frac{1}{\Gamma(2 - \gamma)} \int_0^x \frac{u''(\zeta)}{(t - \zeta)^{\gamma-1}} d\zeta + \omega^\gamma u(t) = 0 \tag{6}$$

where $\omega$ is a positive constant and $1 < \gamma < 2$, which Mainardi called a fractional oscillatory equation. This paper has been, without exaggeration, very interesting for a lot of researchers. First of all, note that:

1.  If $\lambda \neq 0$, then any solution $u(x) \in S^2[0,1]$ (where $S^2[0,1]$ is the class of summable (integrable) on $[0,1]$ functions $u(x)$ including their derivatives of first and second order) for the equation

$$D^{(\beta)} u = \frac{1}{\Gamma(\alpha)} \int_0^x (x - \zeta)^{\alpha-1} u''(\zeta) d\zeta = -\lambda u \tag{7}$$

    coincides with the solution for Equation (4);
2.  Equations (4) and (7) are equal if

$$\lim_{x \to 0} \frac{d^{-(1-\alpha)}}{dx^{-(1-\alpha)}} u(x) = 0.$$

Of course, the fractional oscillatory equation, or equation for fractional oscillator (as an equation, which describes an oscillatory physical system), will have at least the main oscillatory properties.

Hereafter, the following integral equations will play the main role:

$$u(x) - \frac{\lambda}{\Gamma(2 - \alpha)} \left[ \int_0^1 x(1 - \zeta)^{1-\alpha} u(\zeta) d\zeta - \int_0^x (x - \zeta)^{1-\alpha} u(\zeta) d\zeta \right] =$$

$$= u(x) - \frac{\lambda}{\Gamma(2-\alpha)} \int_0^1 G_0(x,\zeta) u(\zeta) d\zeta = 0,$$

$$u(x) - \frac{\lambda}{\Gamma(2-\alpha)} \left[ \int_0^1 x^{1-\alpha}(1-\zeta)^{1-\alpha} u(\zeta) d\zeta - \int_0^x (x-\zeta)^{1-\alpha} u(\zeta) d\zeta \right] =$$

$$u(x) - \frac{\lambda}{\Gamma(2-\alpha)} \int_0^1 G_1(x,\zeta) u(\zeta) d\zeta = 0$$

where $G_0(x,\zeta)$ is the Green's function of the problem (4) and (5), which was constructed in [2] and $G_1(x,\zeta)$ is the Green function of the problem

$$L(u; 1, 1-\alpha, 1, 0) = \frac{1}{\Gamma(\alpha)} \frac{d}{dx} \int_0^x \frac{u'(\zeta)}{(x-\zeta)^{1-\alpha}} d\zeta + \lambda u(x) = 0, \tag{8}$$

$$u(0) = 0, u(1) = 0, \tag{9}$$

which was considered in [5] for the first time (see also references therein).

　　**Important note**: the operators $L(u; 1, 1-\alpha, 1, 0)$ and $L(u; 1, 1, 1-\alpha, 0)$ have the same orders, but the $\gamma_0, \gamma_1, \gamma_2$, of those orders are different.

　　It is easy to show [6] that $G_0(x,\zeta)$ is not with a fixed sign, and this fact says that Equation (7) was incorrectly chosen as the oscillatory equation. Physically, it is clear that the order of the operator $L(u; \gamma_0, \gamma_1, \gamma_2, q(x))$ is close to 2 (or when $\gamma_0 + \gamma_1 + \gamma_2 - 1$ is close to 2), then the operator $L(u; \gamma_0, \gamma_1, \gamma_2, q(x))$ has the main oscillatory properties. We have the following result.

**Theorem 1.** *If*

$$0 < \alpha < \left( \frac{32\pi^2}{9} + \frac{2}{3} \right)^{-1},$$

*then the first eigenvalue of the problem (4) and (5) is positive and simple (the multiplicity of this eigenvalue is equal to 1), and basic (main) tone has no nodes (i.e., the first eigenfunction corresponding to the first eigenvalue does not vanish in* $(0, 1)$*).*

**Proof.** That the first eigenvalue of the problem (4) and (5) is positive and simple for

$$0 < \alpha < \left( \frac{32\pi^2}{9} + \frac{2}{3} \right)^{-1}$$

was proved in [7,8]. We show now that basic (main) tone of the problem (4) and (5) has no nodes.

　　It is known that a number $\lambda$ will be an eigenvalue of the problem (4) and (5) [4] if and only if this value $\lambda$ is the root (zero) of the function $E_{1/\beta}(-\lambda; 2)$ and the corresponding eigenfunctions of the problem (4) and (5) are

$$u_n(x) = x E_{\frac{1}{\beta}}(-\lambda_n x^\beta; 2), n = 1, 2, 3\ldots$$

where

$$\lambda_1, \lambda_2, \ldots, \lambda_n, \ldots$$

are zeros of the function $E_\beta(-\lambda; 2)$, numbered according to the non-decreasing of their modules,

$$E_\rho(z; \mu) = \sum_{k=0}^\infty \frac{z^k}{(\mu + k\rho^{-1})}$$

is a Mittag–Leffler function. We shall show that the function

$$u_1(x) = xE_{\frac{1}{\beta}}(-\lambda_1 x^\beta; 2)$$

does not vanish in $(0,1)$. Let $x_0 \in (0,1)$ be such that

$$x_0 E_{\frac{1}{\beta}}(-\lambda_1 x_0^\beta; 2) = 0.$$

Then the number $\lambda_1 x_0^\beta$ is a zero of $E_\beta(-\lambda; 2)$, moreover $\lambda_1 x_0^\beta < \lambda_1$ (since $x_0 \in (0,1)$). This contradicts the assumption that $\lambda_1$ is the first zero of the function $E_{1/\beta}(-\lambda; 2)$. Theorem 1 is proved. □

Since for $\alpha > \frac{2}{3}$, the function $E_{1/\beta}(-\lambda; 2)$ has no real zeros [9], then the problem (4) and (5) has this main (at least first eigenvalues are real) oscillatory property only for small $\alpha$.

Next, we consider in detail the function $G_1(x, \zeta)$. As it was shown in [7], this function has many useful properties, in particular $G_1(x, \zeta) = G_1(1 - \zeta, 1 - x)$ and $G_1(x, \zeta) > 0$, for any $x, t \in (0,1)$ (i.e., this Green's function is a persymmetric function). Part of the results of the theorem below follow from the well-known Perron's theorem.

**Theorem 2.** *The first eigenvalue $\lambda_1$ of the problem (8) and (9) is real, simple, and satisfies the condition*

$$0 < \lambda_1^{-1} < \frac{\Gamma(2 + 2\alpha)}{\Gamma(1 + \alpha)},$$

*and the basic (main) tone has no nodes for all $\alpha \in (0,1)$.*

**Proof.** As it was written above, from Perron's theorem, it follows that the first eigenvalue is real and simple, and the basic (main) tone has no nodes. Let us show that

$$0 < \lambda_1^{-1} < \frac{\Gamma(2 + 2\tilde{\alpha})}{\Gamma(1 + \tilde{\alpha})}, \quad \tilde{\alpha} = 1 - \alpha$$

holds. As it was mentioned above, the problem (8) and (9) is equivalent to the integral equation of Fredholm (II kind)

$$u(x) + \frac{\lambda}{\Gamma(1 + \tilde{\alpha})} \left[ \int_0^x (x - \zeta)^{\tilde{\alpha}} u(\zeta) d\zeta - x^{\tilde{\alpha}} \int_0^1 (1 - \zeta)^{\tilde{\alpha}} u(\zeta) \, d\zeta \right] = 0,$$

and the value $\lambda$ is an eigenvalue of the problem (8) and (9) if and only if it is a zero of the Mittag–Leffler function $E_{\frac{1}{1+\tilde{\alpha}}}(-\lambda, 1 + \tilde{\alpha})$ [5,7].

Let us rewrite the operator

$$Au = \frac{1}{\Gamma(1 + \tilde{\alpha})} \left[ \int_0^x (x - \zeta)^{\tilde{\alpha}} u(\zeta) d\zeta - x^{\tilde{\alpha}} \int_0^1 (1 - \zeta)^{\tilde{\alpha}} u(\zeta) d\zeta \right]$$

as

$$Au = A_0 u - A_1 u,$$

where

$$A_0 u = \frac{1}{\Gamma(1 + \tilde{\alpha})} \int_0^x (x - \zeta)^{\tilde{\alpha}} u(\zeta) d\zeta,$$

and

$$A_1 u = \frac{1}{\Gamma(1+\tilde{\alpha})} \int_0^1 x^{\tilde{\alpha}}(1-\zeta)^{\tilde{\alpha}} u(\zeta) d\zeta.$$

As operators $A_0$ and $A_1$ are trace class operators [10], then

$$spA = sp(A_0 - A_1) = sp(A_0) - sp(A_1).$$

Since $A_0$ is Volterra's operator, then $sp(A_0) = 0$, and so

$$sp(A) = -sp(A_1).$$

It is easy to find the trace of the operator $A_1$ ($A_1$ it is one-dimensional operator). Let us consider the equation

$$u(x) - \frac{\lambda}{\Gamma(1+\tilde{\alpha})} \int_0^1 x^{\tilde{\alpha}}(1-\zeta)^{\tilde{\alpha}} u(\zeta) d\zeta = 0.$$

The Fredholm determinant of this equation is

$$d(\lambda) = |1 - \lambda K_{11}|,$$

where

$$K_{11} = \frac{1}{\Gamma(1+\tilde{\alpha})} \int_0^1 \zeta^{\tilde{\alpha}}(1-\zeta)^{\tilde{\alpha}} d\zeta = \frac{\Gamma(1+\tilde{\alpha})}{\Gamma(2+2\tilde{\alpha})}.$$

From this we obtain

$$sp(A) = \frac{\Gamma(1+\tilde{\alpha})}{\Gamma(2+2\tilde{\alpha})}.$$

Thus

$$\lambda_1^{-1} + \sum_{i=2}^{\infty} \lambda_i^{-1} = \frac{\Gamma(1+\tilde{\alpha})}{\Gamma(2+2\tilde{\alpha})}.$$

Since the kernel of the operator $-A$ is non-negative, then $\lambda_1$ is a positive number, and $\sum_{i=2}^{\infty} \lambda_i^{-1}$ is positive, so

$$\lambda_1^{-1} < \frac{\Gamma(1+\tilde{\alpha})}{\Gamma(2+2\tilde{\alpha})}.$$

Theorem 2 is proved. □

**Corollary 1.** *Since $\lambda$ is an eigenvalue of the problem (8) and (9) [5,7] if and only if $\lambda$ is a zero of the function $E_{1/(1+\tilde{\alpha})}(-\lambda; 1+\tilde{\alpha})$, and the corresponding eigenfunctions of the problem (8) and (9) are*

$$u_n(x) = x^{\tilde{\alpha}} E_{1/(1+\tilde{\alpha})}(-\lambda_n x^{1+\tilde{\alpha}}; 1+\tilde{\alpha})$$

*$n = 1, 2, 3...$, where $\lambda_1, \lambda_2, ..., \lambda_n, ...$ are zeros of the function $E_{1/(1+\tilde{\alpha})}(-\lambda; 1+\tilde{\alpha})$, numbered by their non-decreased modules, then the function $E_{1/(1+\tilde{\alpha})}(-\lambda; 1+\tilde{\alpha})$ has positive and simple first zero, and the function*

$$u_1(x) = x^{\tilde{\alpha}} E_{1/(1+\tilde{\alpha})}(-\lambda_1 x^{1+\tilde{\alpha}}; 1+\tilde{\alpha})$$

*does not vanish in $(0, 1)$.*

**Conclusion.** From these theorems follows

(a) Which of Equations (4) or (8) is the correct choice as an oscillatory;
(b) How the spectral structure of $L(u; \gamma_0, \gamma_1, \gamma_2, q(x))$ depends **on generatrices** $\gamma_0, \gamma_1, \gamma_2$ of order of fractional differential equation;
(c) How the nature of the modeling process helps to indentify generatrices $\gamma_0, \gamma_1, \gamma_2$ of order of operator.

*2.2. On Completeness of System of Eigenfunctions and Associated Functions of Operator, Generated by Model Fractional Differential Equation and Boundary Conditions of Sturm–Liouville Type*

Let us start from the equation

$$D^{(\sigma_2)}u - [\lambda + q(x)]u(x) = 0, \tag{10}$$

where

$$D^{(\sigma_2)}u = \frac{1}{\Gamma(1-\gamma)}\frac{d}{dx}\int_0^x \frac{u'(\zeta)}{(x-\zeta)^\gamma}d\zeta, \ 0 < \gamma < 1, \ \sigma_2 = 1 + \gamma.$$

At first, Equation (10) was studied in [5] as a model equation of the fractional order $1 < \sigma_2 < 2$. In particular, it was established in [5] that the two-point Dirichlet problem

$$u(0) = 0, \quad u(1) = 0, \tag{11}$$

for Equation (10) with $q(x) = 0$ is equivalent to the integral equation

$$\frac{1}{\Gamma(2-\gamma)}\left[\int_0^x (x-t)^{1-\gamma}u(t)dt - \int_0^1 x^{1-\gamma}(1-t)^{1-\gamma}u(t)dt\right] = \lambda u.$$

We have:

**Theorem 3.** *Let* $\gamma_0 = \gamma_1 = 1, q(x) \equiv 0$. *Then the system of eigenfunctions and associated functions of the problem (10) and (11) is complete in* $L_2(0,1)$.

A close result (for a semibounded potential $q(x)$) was obtained in [11]. It should be noted that the proof of these statements are based on the fact that the operator, generated by the problem (10) and (11), is sectorial [12].

**Theorem 4.** *All eigenvalues of the problem (10) and (11) for* $q(x) \equiv 0$ *are in the angle* $|arg z| < \frac{\pi(1-\gamma)}{2}$, $0 < \gamma < 1$.

**Proof.** Consider the expression $(-D^{(\sigma_2)}f, f)$. It is obvious that

$$(-D^{(\sigma_2)}f, f) = -\left(\frac{1}{\Gamma(1-\gamma)}\frac{d}{dx}\int_0^x \frac{f'(t)}{(x-t)^\gamma}dt, f(x)\right)$$

$$= (\frac{1}{\Gamma(1-\gamma)}\int_0^x \frac{f'(t)}{(x-t)^\gamma}dt, f'(x)) = (_0J_x^\alpha f', f')$$

where $\alpha = 1 - \gamma$ and $_0J_x^\alpha$ is the operator of fractional integration of order $\alpha$:

$$(_0J_x^\alpha f)(x) = \frac{1}{\Gamma(\alpha)}\int_0^x (t-s)^{1-\alpha}f(s)ds.$$

By a well-known Matsaev–Palant theorem ([13], p. 481), the values of the form $({}_0J_x^{\alpha}f', f')$ is in the angle $|arg z| < \frac{\pi \alpha}{2}$. This proves Theorem 4. $\square$

Since the number $\lambda$ is an eigenvalue of the problem (10) and (11) if $\lambda$ is a zero of the function $E_{1/\mu}(-\lambda; \mu)$ ($\mu = 1 + \gamma$) [5], the following proposition is valid.

**Corollary 2.** *All zeros of the function $E_{1/\mu}(-\lambda; \mu)$ are in the angle $|arg z| < (\pi(1-\gamma))/(2), 0 < \gamma < 1$. Here $\mu = 1 + \gamma$.*

**Theorem 5.** *The problem (10) and (11) for $q(x) \equiv 0$ has no eigenvalues inside the circle with radius $\Gamma(4 - 2\gamma)/\Gamma(2 - \gamma)$ centered at the coordinate origin.*

*2.3. Methods of the Theory of Perturbations in Fractional Calculus and the Questions of Localization and Multiplicity of Eigenvalues*

To prove that the studied operator does not have the associated functions, we present the main points of the method presented in [14].

In $L^2(0, 1)$ we consider the operator

$$A_\rho(u) = \int_0^1 G(x, t)u(t)\, dt = \frac{1}{\Gamma(\rho^{-1})} \left[ \int_0^x (x - t)^{\frac{1}{\rho}-1} u(t) dt - \int_0^1 x^{\frac{1}{\rho}-1}(1 - t)^{\frac{1}{\rho}-1} u(t)\, dt \right],$$

which was for the first time studied in [7]. Here, $0 < \rho < 2$, and

$$G(x, t) = \begin{cases} \dfrac{(1 - t)^{\frac{1}{\rho}-1} x^{\frac{1}{\rho}-1} - (x - t)^{\frac{1}{\rho}-1}}{\Gamma(\rho^{-1})}, & 0 \le t \le x \le 1, \\[2mm] \dfrac{(1 - t)^{\frac{1}{\rho}-1} x^{\frac{1}{\rho}-1}}{\Gamma(\rho^{-1})}, & 0 \le x \le t \le 1, \end{cases}$$

is the Green function of the following problem $S$ (for $\lambda = 0$):

$$\frac{1}{\Gamma(n - \rho^{-1})} \frac{d^n}{dx^n} \int_0^x (x - s)^{n - \rho^{-1} - 1} u(s) ds + \lambda\, u = 0,$$

($n - 1 \le \rho^{-1} < n$, $n = [\rho^{-1}] + 1$, where $[\rho^{-1}]$ is the integer part of the number $\rho^{-1}$)

$$u(0) = 0, u'(0) = 0, \cdots, u^{(n-2)}(0) = 0, u(1) = 0.$$

In this case [7], if $\gamma_0 = \gamma_1 = \cdots = \gamma_n = 1$ then the problem $S$ takes the form

$$u^{(n)} + \lambda\, u = 0,$$

$$u(0) = 0, u'(0) = 0, \cdots, u^{(n-2)}(0) = 0, u(1) = 0,$$

of which the Green function $G(x, t)$ (for $\lambda = 0$) reads

$$G(x, t) = \begin{cases} \dfrac{(1 - t)^{n-1} x^{n-1} - (x - t)^{n-1}}{(n - 1)!}, & 0 \le t \le x \le 1, \\[2mm] \dfrac{(1 - t)^{n-1} x^{n-1}}{(n - 1)!}, & 0 \le x \le t \le 1. \end{cases}$$

The last function was studied very well, and we will use it in the sequel. The operator $A_\rho$ was investigated in [5,7,15]. Let us study this operator carefully, because it turns out that the Mainardi equation [3] (fractional oscillatory equation) does not have many basic oscillatory properties. The

search for such a differential equation that has these properties led us to study the operator $A_\rho$. Now, let us introduce the most significant properties of this operator established by the author earlier:

1. For $\rho > 1$ the operator $A_\rho$ is completely nonself-adjoint [5,7,15];
2. For $\rho \leq 1$ the operator $A_\rho$ is sectorial [6] (and see the references therein);
3. For $0 < \rho < 2$ the system of eigenfunctions of the operator $A_\rho$ is complete in $L_2(0,1)$ [5,16];

Now, let us study integral operators corresponding to boundary value problems for fractional differential equations using methods of the theory of perturbations.

The holomorphic dependence of these operators on the order of fractional differentiation is proved. There are several useful criteria for holomorphy. In accordance with this, various types of holomorphic families are considered. We will use type $(A)$. Type $(A)$ is defined in terms of the boundedness of the perturbation with respect to the unperturbed operator.

Let us formulate a very important criterion, which we will use later [17].

**Theorem 6.** *(Criterion of holomorphy* $(A)$*). Let T be a closable operator from X in Y, and let* $T^{(n)}$*,* $n = 1, 2, ...,$ *be operators from X in Y, of which the domains of definition contain* $D(T) = D$*. Assume that there exists constants* $a, b, c \geq 0$*, such that*

$$T^{(n)}u \leq c^{n-1}(a||u|| + b||Tu||),\ u \in D,\ n = 1, 2, .... \tag{12}$$

*Then for* $|\kappa| < 1/c$ *the series*

$$T(\kappa)u = Tu + \kappa T^{(1)}u + \kappa^2 T^{(2)}u + ...,\ u \in D$$

*defines the operator* $T(\kappa)$ *with the domain of definition D. If* $|\kappa| < (b+c)^{-1}$*, then the operator* $T(\kappa)$ *is closable, and the closures* $\widetilde{T}(\kappa)$ *form a holomorphic family of type* $(A)$ *[7].*

We shall note that the holomorphic families of this type and, in particular bounded-holomorphic families, were studied since Rellich's papers [18] (and references therein). A wide list of references is presented in papers of M.K. Gavurin and V.B. Loginov [18] (and references therein).

**Theorem 7.** *If* $|\varepsilon| < 1$*, then the operator*

$$A(\varepsilon)u = -\int_0^x (x-t)^{1+\varepsilon}u(t)dt + \int_0^1 x^{1+\varepsilon}(1-t)^{1+\varepsilon}u(t)dt$$

*forms a holomorphic family of type* $(A)$*, i.e.,*

$$A(\varepsilon)u = A(0)u + \varepsilon A_1 u + \varepsilon^2 A_2 u + ... + \varepsilon^n A_n u + ...$$

*where*

$$A(0)u = -\int_0^x (x-t)u(t)dt + \int_0^1 x(1-t)u(t)dt$$

*the unperturbed operator, and*

$$A_n u(x) = \int_0^x \left( \widetilde{K}(x,t)_n - K(x,t)_n \right) u(t)dt,$$

$$\widetilde{K}(x,t)_n = \frac{x(1-t)\ln^n(1-t)x}{n!},$$

$$K(x,t)_n = \begin{cases} \frac{x(1-t)\ln^n(1-t)x}{n!}, t < x, \\ 0, t \ge x. \end{cases}$$

**Theorem 8.** *If $|\varepsilon| < 3/2$, then the operator*

$$\widetilde{B}(\varepsilon)u = -\int_0^x (x-t)^{1+\varepsilon}u(t)dt + \int_0^1 x(1-t)^{1+\varepsilon}u(t)dt$$

*forms a holomorphic family of type $(A)$ where*

$$B(0)u = -\int_0^x (x-t)u(t)dt + \int_0^1 x(1-t)u(t)dt$$

*is the unperturbed operator, and*

$$B_n u(x) = \int_0^x \left(\overline{K}(x,t)_n - K(x,t)_n\right) u(t)dt,$$

*where*

$$\overline{K}(x,t)_n = \frac{x(1-t)\ln^n(1-t)x}{n!},$$

*and*

$$K(x,t)_n = \begin{cases} \frac{x(1-t)\ln^n(1-t)x}{n!}t < x, \\ 0, t \ge x. \end{cases}$$

Since [15] the Fredholm spectrum of the operators under study coincides with the zeros of the appropriate function of the Mittag–Leffler type, the presented method allows to efficiently study the problem of distribution of zeros for functions of Mittag–Leffler type. To confirm this assertion, we give two examples. Following [8], we introduce the following notation: $\lambda_n(\alpha)$ are the eigenvalues of the problem (4) and (5). In [8], it was written that "... in the limiting case $\alpha = 0$ the problem (4)–(5) becomes the Sturm–Liouville boundary value problem with the sequence of eigenvalues $\lambda_n(\alpha) = (\pi n)^2$. Is it true that $\lim_{\alpha \to 0+} \lambda_n(\alpha) = (\pi n)^2$ for any fixed $n$? The answer will be positive."

Let us prove a stronger proposition.

**Theorem 9.** $\lim_{\alpha \to \alpha_0+} \lambda_n(\alpha) = \lim_{\alpha \to \alpha_0-} \lambda_n(\alpha) = \lambda_n(\alpha_0)$ *for any $\alpha_0 \in [0,1]$.*

**Proof.** Theorem 8 is a trivial corollary of Theorem 4.2 (see [10], p. 35) and the fact that the operator function $\widetilde{B}(\varepsilon)$ is strongly continuous for $|\varepsilon| < 1$. □

Finally, we consider one more significant question of the multiplicity of eigenvalues of the operator $\widetilde{B}(\varepsilon)$ (as was mentioned above, this question is related to the question of the multiplicity of zeros of a corresponding function of the Mittag–Leffler type [7]).

It is known ([1], theorem of Dzhrbashian–Nersesian) that all zeros of a function of the Mittag–Leffler type $E_\rho(z, \mu)$ (where $\rho > 1/2$, $\rho \ne 1$, $Im(\mu) = 0$) that are sufficiently large in the modulus are simple. Therefore, we mainly pay attention to the multiplicity of the first eigenvalues of the operator $\widetilde{B}(\varepsilon)$. The following theorem holds [14]

**Theorem 10.** *Let $|\varepsilon| < \left(\frac{32\pi^2}{9} + \frac{2}{3}\right)^{-1}$. Then the first eigenvalue $\lambda_1(\varepsilon)$ of the operator $\widetilde{B}(\varepsilon)$ is simple [7].*

**Proof.** It is known [7] that if the spectrum of the operator $\widetilde{B}(0)$ is divided into two parts by a closed curve $\Gamma$, then the spectrum of the operator $\widetilde{B}(\varepsilon)$ is also divided by the curve $\Gamma$ for sufficiently small $\varepsilon$. In this case, the estimate of the smallness of $\varepsilon$ is as follows [7]:

$$|\varepsilon| < min_{\zeta \in \Gamma}(a||R(\zeta, \widetilde{B}(0))|| + b||\widetilde{B}(0)R(\zeta, \widetilde{B}(0))|| + c)^{-1} \tag{13}$$

(where $a$, $b$, and $c$ are parameters that enter inequality (12)). As the contour $\Gamma$ in Formula (13), we take the circumference $|\zeta - \frac{1}{\pi^2}| = \frac{\rho}{2}$, where $\rho$ is the distance from $\frac{1}{\pi^2}$ to the set of the rest eigenvalues of the operator $\widetilde{B}(0)$. The parameters $a, b$ and $c$ are already calculated [7]. Theorem 9 is proved. $\square$

We note that it can be shown in the same way that the second eigenvalue of the operator $\widetilde{B}(\varepsilon)$ is simple too. It is the principal point that this method gives the possibility to include the study of nonselfadjoint operators of the form $A_\gamma^{[\alpha,\beta]}$ (and not only operators of the form $A_\gamma^{[\alpha,\beta]}$) in the general scheme of perturbation theory.

*2.4. Solving the Problem of Finding the Radon Flux Density by Its Concentration at Different Depths of the Earth's Surface by the Method of Separated Variables*

In the last few years, fractional integro-differentiation has been the focus of many researchers of science and engineering [19,20] (and see references therein). We can describe anomalous diffusion or dispersion using the fractional space derivatives, and some processes with 'memory' effects using the fractional time derivatives. In this paragraph, we solve the problem of finding the radon flux density [14] by its concentration at different depths of the earth's surface by the method of approximate solution of the first boundary value problem for the fractional differential equation of advection-diffusion [21].

$$\frac{\partial u(x,t)}{\partial t} = D_{0+}^\alpha u(x,t).$$

It is known [21,22] that the problem of finding the radon flux density by its concentration at different depths of the earth's surface is set as follows: to find a solution to the boundary value problem

$$\frac{\partial u(x,t)}{\partial t} = D_{0+}^\alpha u(x,t),$$

where $D_{0+}^\alpha u(x,t)$—the Riemann-Liouville fractional derivative of the order $\alpha$, with boundary conditions

$$u(0,t) = u(1,t) = 0, \tag{14}$$

$$u(x,0) = \phi(x), \tag{15}$$

Using the method of separation of variables [21], we can write out the solution to this problem

$$u(x,t) = \sum_{n=1}^{\infty} \delta_n exp(\lambda_n t)x^{\alpha-1}E_{\alpha,\alpha}(\lambda_n x^\alpha). \tag{16}$$

here

$$..., \lambda_{-3}, \lambda_{-2}, \lambda_{-1}, \lambda_1, \lambda_2, \lambda_3, ...$$

- Zeros of the function $E_{\alpha,\alpha}(\lambda)$, arranged in the appropriate order according to [21], and

$$\delta_n = \{\delta(x), z_n(x)\}_{L_2(0,1)}, \ n = 1, 2, ...$$

- Fourier coefficients $\phi(x)$, and the system of functions $z_n{}_{n=1}^{\infty} = (1-x)^{\alpha-1}E_{\alpha,\alpha}(\lambda_n(1-x)^{\alpha})$ is the biorthogonal system of eigenfunctions $\omega_n(x) = x^{\alpha-1}E_{\alpha,\alpha}(\lambda_n x^{\alpha})$ ($z_n$—is the system of eigenfunctions of the contiguous boundary problem).

For an approximate solution of this problem, one can use the formula

$$u(x,t) \approx \sum_{n=1}^{N} \delta_n exp(\lambda_n t)x^{\alpha-1}E_{\alpha,\alpha}(\lambda_n x^{\alpha}).$$

In [23], using this formula, the problem of finding the radon flux density by its concentration at different depths of the earth's surface was solved. It was shown that

$$u(x,t) \approx \sum_{n=1}^{50} \delta_n exp(\lambda_n t)x^{\alpha-1}E_{\alpha,\alpha}(\lambda_n x^{\alpha}).$$

rather well approximates the exact solution $u(x,t)$ and also there are algorithms for finding the eigenvalues $\lambda_n$ and the Fourier coefficients $\delta_n$.

**Remark 1.** *Note the paper [24], where the same problem is solved by numerical methods.*

### 3. Method of Separation of Variables for Time–Space Fractional Vibration Equations—The Basic Theory

In this paragraph we present the necessary information from the spectral theory of operators generated by differential equations of the second order with fractional derivatives in the lowest terms with boundary conditions of the Sturm–Liouville type.

Many problems of mathematical physics [25–27] associated with perturbations of normal operators with discrete spectrum lead to the consideration in Hilbert space $\mathfrak{H}$ of the compact operator

$$A = (I+S)H,$$

called *a weak perturbation H* (for a compact S) or as the operator of *Keldysh type* (the information about of such operators and last investigations in this field were published in our brief [28]).

In [28] the basis property of the system of root vectors and localization of root vectors and eigenvalues for the investigated operators were established (we shall also note paper [29] of one of our co-authors E. Larionov, in which the spectral theory of the operators of the Keldysh type is very strongly developed).

In the present paper, we consider the operator of Keldysh type $B$, generated by the differential expression

$$u'' + \varepsilon D_{0x}^{\alpha}u = \lambda u, \tag{17}$$

and the boundary conditions of the Sturm–Liouville type

$$u(0) = 0, u(1) = 0. \tag{18}$$

Note, that for $0 < \alpha < 2$, the spectral structure of operator $\tilde{B}$ generated by problem

$$u'' + \varepsilon D_{0x}^{\alpha}u = \lambda u, \tag{19}$$

$$u(0) = 0, u(1) = 0, \tag{20}$$

was considered in detail in our paper [12]. In particular, the following theorem was shown:

**Theorem 11.** *If $|\varepsilon| < \frac{10}{20}$, then all eigenvalues of operator $\tilde{B}$ are simple and real.*

From this theorem, it follows that the operator $\tilde{B}$ generated no associated functions.

**Theorem 12.** *The number $\lambda$ is an eigenvalue of problem (19) and (20) if $\lambda$ is a zero of the function*

$$\omega(\lambda) = 1 + \sum_{n=1}^{\infty} \sum_{m=0}^{n} (-\varepsilon)^n \frac{C_n^m \lambda^{n-m}}{\Gamma(2n - m\alpha + 2)}. \tag{21}$$

The eigenfunctions of problem $(A)$ take the form

$$\chi_i(x) = x + \sum_{n=1}^{\infty} \sum_{m=0}^{n} (-\varepsilon)^n \frac{C_n^m \lambda_i^{n-m}}{\Gamma(2n - m\alpha + 2)} x^{2n-m\alpha+1}, \tag{22}$$

where $\lambda_i$ are zero of the function $\omega(\lambda)$. In [12] it was proved that the system of eigenfunctions (22) is complete in $L_2(0,1)$. However, this system is not orthogonal. Therefore, in paper [12] they were considered together with problems (19) and (20), the problem conjugate to it.

Let us consider operator $B$, generated by the differential Equation (19) and boundary conditions (20). The following theorem holds [14]

**Theorem 13.** *Let $0 < \alpha < 2$, then, the system of eigenfunctions of operator $B$ is complete in $L_2(0,1)$.*

**1.** Next, let us denote $n(r,B)$ the exact number of characteristic values of the operator $B$ lying in circle $|\lambda| \leq r$. The problem of allocation of characteristic values of the operator $B$ formulates an investigation of asymptotic properties of $n(r,B)$ for $r \to \infty$. In [25], this problem was solved when the order of fractional derivative $D_{0x}^{\alpha}$ was less than 1. In [25], the study of the function $n(r,B)$ was reduced to the one of the spectra for the linear beam operator $L(\lambda) = I + M - \lambda N$.

Since $M$ is a compact operator and $N$ is a positive operator, then by Keldysh's theorem ([10], p. 318) we have

$$\lim_{x \to 0} \frac{n(r,B)}{n(r,N)} = 1$$

if for the distribution function $n(r,B)$ of characteristic values of the operator $N$ we may choose non-decreasing function $\varphi(r)(0 \leq r \leq \infty)$ such that [10]:

1. $\lim\limits_{r \to \infty} \varphi(r) = \infty$;
2. $\lim\limits_{r \to \infty} (\ln \varphi(r))' < \infty$;
3. $\lim\limits_{r \to \infty} \frac{n(r,B)}{\varphi(r)} = 1$.

Obviously, in our case as in [25] we may take the function $\sqrt{r}$ as $\varphi(r)$. Any linearized mechanical system in which there is energy dissipation is described by a linear operator $A$, densely defined in $H$, with values of the form $(Af, f)$ in the left half-plane:

$$Re(Af, f) \leq 0, \ (f \in D_A).$$

In quantum mechanics, energy dissipation is characterized by the fact that the form of the linear operator describing the physical system lies in the upper half-plane, i.e.,

$$Im(Af, f) \geq 0, \ (f \in D_A).$$

For definiteness, when speaking of dissipative operators, we shall have in mind the operators of the latter type; dissipative operators of quantum mechanics.

**2.** Since (see [14] and references therein) any linearized mechanical system, which has an energy dissipation described by a linear operator $\tilde{A}$, is densely defined in a Hilbert space $\mathfrak{H}$ with values of the form $(\tilde{A}f, f)$ in the left half-plane

$$Re(\tilde{A}f, f) \leq 0, \ (f \in \mathfrak{D}_{\tilde{A}}).$$

As the operator $\tilde{B}$ describes oscillations of the mechanical system, then it should be dissipative (see [30] and references therein).

In this paragraph we show that the operator $\tilde{B}$ is dissipative.

First, we shall note papers of F. Tricomi, Matsaev and Palant [25] (and references therein) (where it was shown, that values of the form $(I^\alpha f, f)$ are lying in the angle $|\arg \lambda| \leq \frac{\alpha \pi}{2}$, here $I^\alpha$—is fractional integral in the Riemann–Liouville sense of order $\alpha$) and papers of authors [25] (and references therein), where it was established

$$Re(D_{0x}^\alpha u, u) \geq 0, 0 < \alpha < 1, \tag{23}$$

and

$$Re(D_{0x}^\alpha u, u) \leq 0, 1 < \alpha < 2. \tag{24}$$

**Theorem 14.** *If $0 < \alpha < 1$ and $\varepsilon > 0$, then the operator, generated by the problem*

$$u'' - \varepsilon D_{0x}^\alpha u = \lambda u$$

$$u(0) = 0, \ u(1) = 0$$

*is dissipative.*

**Proof.** This theorem follows from the relation (23) and the fact that the operator

$$Tu = \begin{cases} -u'', \\ u(0) = 0, \ u(1) = 0, \end{cases}$$

is dissipative. $\square$

**Theorem 15.** *If $1 < \alpha < 2$ and $\varepsilon < 0$, then the operator, generated by the problem*

$$u'' - \varepsilon D_{0x}^\alpha u = \lambda u$$

$$u(0) = 0, \ u(1) = 0$$

*is dissipative.*

**Proof.** The scheme of the proof of Theorem 14 is the same as the one of Theorem 13. $\square$

**Remark 2.** *Let $D = \{0 < x < 1, 0 < t < 1\}$, and consider the first boundary value problem for equations of vibration of a string with a fractional derivative of order $\alpha$ with respect to partial variable*

$$\frac{\partial^2 u}{\partial t^2} = \frac{\partial^2 u}{\partial x^2} + C_1 D_{0x}^\alpha u + C_0 D_{0t}^\beta u, \ 0 < \alpha, \ \beta < 2, \tag{25}$$

$$u(0, t) = u(1, t), \tag{26}$$

$$u(x, 0) = \varphi(x), \tag{27}$$

$$u_t'(x, 0) = \varphi(x), \tag{28}$$

If $C_0$ is negative, the numerical methods can work well, and the related numerical theoretical results can also be well established. However, when $C_0$ is positive, our numerical methods can not compute well, and the convergence and stability of the proposed methods can not be proved. This is a consequence of the fact that the term $C_0D^\beta$ describes dissipation, as it was established by the Theorems 13 and 14 (in the case when $C_0$ is negative, the physical sense of the term $C_0D^\beta$ is incomprehensible). Thus, proved Theorems 13 and 14 allow to correctly formulate the boundary value problem for Equation (25).

**3.** Let us consider operator $A$, generated by the problem

$$u'' - \varepsilon D_{0x}^\alpha u = \lambda u, \tag{29}$$

$$u(0) = 0, \; u(1) = 0, \tag{30}$$

where $0 < \alpha < 2$.

Finally, let us show that operator $A$ is oscillatory (if the operator describes the oscillation motions, then it should have a whole complex of the oscillatory properties).

It is known that [25] (and references therein) if $0 \leq \varepsilon \leq \frac{1}{3}$, and $1 < \alpha < 2$, then the Green function of the problem (29) and (30) is of fixed sign (we shall note, that Green's function of problem (29) and (30) was firstly constructed by one of the authors in his paper [25] (and references therein)). Unfortunately, this very important property of Green's function is possible to get only for a small enough $\varepsilon$. This is primarily due to the fact that Green's function $G_2(x, \tau)$ [4] of the problem (29) and (30), for $1 < \alpha < 2$, has the following complex structure

$$G_2(x, \tau) = G_1(x, \tau) - \frac{\varepsilon}{E_{1/2}(\varepsilon, 2)} \int_\tau^1 E_\beta[\varepsilon(\eta - \tau)]^\beta d\eta \int_0^1 G(x, t) D_{0t}^{\alpha-1} E_\beta[\varepsilon t^\beta] dt,$$

$$G_1(x, \tau) = \begin{cases} (1-x)\int_\tau^x E_\beta[\varepsilon(t - \tau)]dt - \\ -x - \int_x^1 E_\beta[\varepsilon(t-\tau)^\beta]dt, x \geq \tau, \\ -x \int_\tau^1 E_\beta[\varepsilon(t-\tau)^\beta]dt, x \leq \tau. \end{cases}$$

For $0 < \alpha < 1$, $|\varepsilon| < 1/4$, Green's function of the problem (29) and (30) was constructed in [25] (and references therein). Let us show how this function was constructed. Since the problem (29) and (30), for $0 < \alpha < 1$, is equivalent to the equation

$$u(x) + \frac{\varepsilon}{\Gamma(2-\alpha)}\left\{\int_0^x (x-t)^{1-\alpha}u(t)dt - \int_0^1 x(1-t)^{1-\alpha}u(t)dt\right\} = \lambda \int_0^1 G(x, t)u(t)dt$$

then

$$u(x) = \lambda(I - \varepsilon K)^{-1}\int_0^1 G(x, t)u(t)dt,$$

where

$$G(x, t) = \begin{cases} t(x-1), t \leq x, \\ x(t-1), t > x, \end{cases}$$

$$Ku = -\frac{1}{\Gamma(2-\alpha)}\int_0^x (x-t)^{1-\alpha}u(t)dt + \frac{1}{\Gamma(2-\alpha)}\int_0^1 x(1-t)^{1-\alpha}u(t)dt$$

$$= \left( x J_{0,1}^{2-\alpha} - J_{0,x}^{2-\alpha} \right) u.$$

We can show that

$$K^2 u = K \cdot Ku = (Kx) J_{0,1}^{2-\alpha} u - x J_{0,1}^{4-2\alpha} u + J_{0,x}^{4-2\alpha} u$$

$$K^3 u = K \cdot K^2 u = (K^2 x) J_{0,1}^{2-\alpha} u - (Kx) J_{0,1}^{4-2\alpha} u + x J_{0,1}^{6-3\alpha} u - J_{0,x}^{6-3\alpha} u.$$

By induction, we have

$$K^n = \sum_{i=1}^{n} (-1)^{i+1} (K^{n-i} x) J_{0,1}^{(2-\alpha)i} u + (-1)^{n+2} J_x^{(2-\alpha)n} u.$$

Thus,

$$(I - \varepsilon K)^{-1} = I + \sum_{n=1}^{\infty} (\varepsilon K)^n u$$

$$= I + \sum_{n=1}^{\infty} (\varepsilon)^n \left[ \sum_{i=1}^{n} (-1)^{i+1} (K^{n-i} x) J_{0,1}^{(2-\alpha)i} u + (-1)^{n+2} J_x^{(2-\alpha)n} u \right]$$

$$= I + \sum_{n=1}^{\infty} \sum_{i=1}^{n} (\varepsilon)^n (-1)^{i+1} (K^{n-i} x) J_{0,1}^{(2-\alpha)i} u + \sum_{n=1}^{\infty} (-1)^{n+2} J_x^{(2-\alpha)n} u.$$

Since, for $|\varepsilon| < 1/4$ the kernel $k(x, t)$ of the operator $K$ satisfies the condition

$$|k(x, t)| < 2,$$

we have that the Green's function of problem (29)-(30) is of fixed-sign for $0 < \alpha < 1$ too.

Note the paper [29], which is very important in our opinion, which contains the proof of the basis property for the eigenfunctions.

### 3.1. Parametric Identification for Time–Space Fractional Vibration Equations

In numerous publications of the last decades, the problem of identifying the parameters of fractional models is mainly solved at a theoretical level, for example, by methods of spectral analysis. In our paper [31] (and see references therein), the model parameters are determined based on several characteristic points obtained in the experiment, by substituting the deformation values in the analytical solutions of the corresponding problem. We will use the same technique in what follows to identify the order of the fractional derivative in model (1).

### 3.1.1. The Bagley–Torvik Equation and the Laplace Transform

We consider the problem

$$u''(x) + cD^\alpha u(x) + \lambda u(x) = 0, \ u(0) = 0, \ u'(x) = 1, \tag{31}$$

where $D^\alpha u(x)$ is a fractional derivative of the order $\alpha$. When $1 < \alpha < 2$, by the Riemann–Liouville definition, this problem is presented as follows [32]:

$$D^\alpha u(x) = \frac{d^2}{dx^2} \left[ \frac{1}{\Gamma(2-\alpha)} \int_0^t \frac{u(\tau)}{(x-\tau)^{\alpha-1}} d\tau \right].$$

Equation (31) was proposed in papers [33,34] (and see references therein) for modeling the oscillatory properties of various viscoelastic materials (polymers, glass, etc.). We shall note one recent paper [35] (and see references therein) where this scheme is used to model changes of the stress–strain

characteristics of polymer concrete when subjected to loadings. In these papers, the polymer concrete samples based on polyester resin (dian- and dihloangidrid-1,1-dichloro-2,2-diethylene) were studied. Polymer concrete is represented as the set of granules of a mineral filler that is located in a viscoelastic medium. In this case, the motion of a granule is described by Equation (31), where $\lambda$ is the rigidity modulus of resin, $\alpha$ is the viscoelasticity parameter of the medium and $c$ is the viscosity modulus of resin.

Note that physical systems modeled by Equation (31) are very sensitive to changes in the order of fractional damping and it lead us to the very important task of the parametric identification of this value. We shall note that the problem of the parametric identification [36] (see and the references therein) remains poorly understood. The paper [37] (see and the references therein) is devoted to solving this important problem. Let us briefly give the technique presented in this paper.

We integrate Equaiton (31) from 0 to $x$ and transform the resulting expression. We have

$$\int_0^x u''(t)dt + c \int_0^x D^\alpha u(t)dt + \lambda \int_0^x u(t)dt = 0,$$

$$\int_0^x du'(t)dt + \frac{c}{\Gamma(2-\alpha)} \int_0^x \frac{d^2}{dt^2}\left[\int_0^t \frac{u(\tau)}{(t-\tau)^{\alpha-1}}\right] + \lambda \int_0^x u(t)dt = 0, \tag{32}$$

$$u'(x) - u'(0) + \frac{c}{\Gamma(2-\alpha)}\frac{d}{dt}\left[\int_0^t \frac{u(\tau)d\tau}{(t-\tau)^{\alpha-1}}\right]\Big|_0^x + \lambda \int_0^x u(t)dt = 0,$$

$$u'(x) - 1 + \frac{c}{\Gamma(2-\alpha)}\frac{d}{dt}\left[\int_0^x \frac{u(t)dt}{(x-t)^{\alpha-1}}\right] + \lambda \int_0^x u(t)dt = 0.$$

Obtained expression (32) we integrate from 0 to $x$ again:

$$\int_0^x u'(t)dt - \int_0^x dt + \frac{c}{\Gamma(2-\alpha)}\int_0^x \frac{d}{dt}\left[\int_0^t \frac{u(\tau)d\tau}{(t-\tau)^{\alpha-1}}\right] + \lambda \int_0^x \int_0^t u(\tau)d\tau dt = 0, \tag{33}$$

$$u(x) - x + \frac{c}{\Gamma(2-\alpha)}\int_0^x (x-t)^{1-\alpha}u(t)dt + \lambda \int_0^x \int_0^t u(\tau)d\tau dt = 0.$$

We solve the latest Equation (33) using the Laplace transform. Let us designate by $U(s)$ the image of the function $u(x)$; i.e., $U(s) = u(t)$ or [38] (which is the same),

$$U(s) = \int_0^\infty e^{-st}u(t)dt.$$

It is clear that the function

$$\int_0^x (x-t)^{1-\alpha}u(t)dt$$

represents the convolution of the functions $u(x)$ and $x^{1-\alpha}$. For the convolution of functions, there exists the simple formula of images

$$\int_0^x f_1(x)f_2(x-t)dt = F_1(s)F_2(s),$$

where $F_1(s)$ and $F_1(s)$ are the images of the functions $f_1(x)$ and $f_2(x)$, respectively.

It is clear that

$$\int\limits_0^x \int\limits_0^t u(\tau)d\tau dt = \int\limits_0^x dt \int\limits_0^t u(\tau)d\tau = U(s)/s^2. \tag{34}$$

After some clearly transformations we obtain

$$U(s) - \frac{1}{s^2} + cU(s)s^{\alpha-2} + \lambda U(s)/s^2 = 0. \tag{35}$$

From this we obtain the formula for the image

$$U(s) = \frac{1}{s^2 + cs^\alpha + \lambda}. \tag{36}$$

Formula (36) makes it possible to express the solution of problem (31) using Laplace's integral

$$u(x) = \frac{1}{2\pi i} \int\limits_{\sigma-i\infty}^{\sigma+i\infty} e^{st} U(s)ds. \tag{37}$$

### 3.1.2. Numerical Construction of the Solution

The obtained Formula (37) allows to numerically construct the graphs of solutions. The calculations are performed using the package Mathcad 14. Figure 1 presents the graphs of solutions for various values of parameter $\alpha$. As in [35] we take $c = 1.8$ and $\lambda = 93$. Note that used parameter values were obtained in the experiments on the samples of polymer concrete [35]. The presented numerical check proves the validity of the limit behavior of the solution, which transforms into harmonic oscillations for values of $\alpha$ that are close to 2.

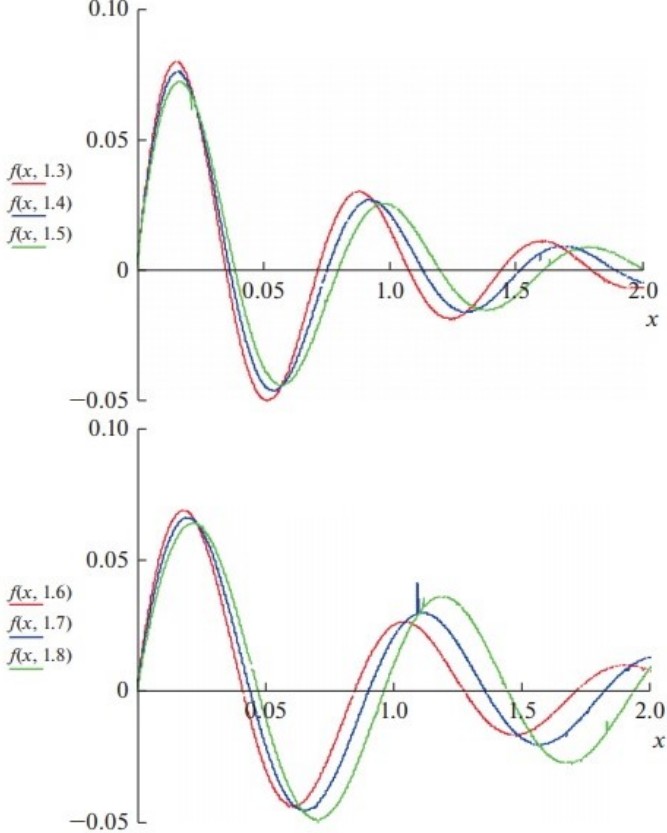

**Figure 1.** Graphs of solutions when $1 < \alpha < 2$.

To prove the correctness of the formulation of the problem of the parametric identification, it is necessary to investigate the solution's stability to the inaccuracy of the parameter $\alpha$ ($\alpha$ has an arbitrary value from (0,1)). For this purpose, in the neighborhood of the point $\alpha$, consider the relative increment of this parameter by $\delta$ (i.e., $\alpha' = \alpha(1 + \delta)$) and determine the deviation function $\rho(\alpha, \delta)$ with respect to the norm in the space $L_1$ (the of the summable functions)

$$\rho(\alpha, \delta) = \int |u(x, \alpha) - u(x, \alpha')| dx, \tag{38}$$

where $u(x, \alpha)$ is the solution of (31) with the parameter $\alpha$. Here, the function $\varepsilon(\alpha, \delta) = \frac{\partial \rho}{\partial \delta}$ determines the sensitivity of the solution to a possible error of the parameter $\alpha$. The values of the function $\varepsilon(\alpha, \delta)$ for various values of parameter $\alpha$ and the levels, 0.1, and 0.15 are found numerically; they are presented graphically in Figure 2.

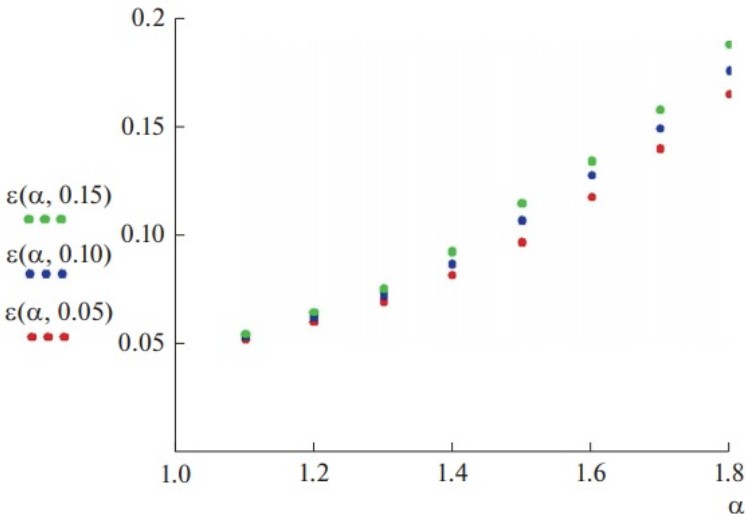

**Figure 2.** On the question of the sensitivity of the solutions of Problem (31) to errors of the parameter.

The obtained values of the function $\varepsilon(\alpha, \delta)$ show the growth of the sensitivity with increasing of parameter $\alpha$. The maximum value of the function $\varepsilon(\alpha, \delta)$ does not surpass 0.2; this allows us to draw a conclusion about the stability of the solutions of problem (31) in relation to a small error of parameter $\alpha$ and the correctness of the problem of identifying this parameter.

It is known that [31] the solution of Cauchy's problem (31) is determined by the following formula

$$u(x) = x - \sum_{n=1}^{\infty} \sum_{m=0}^{n} (-1)^{n+1} \frac{C_n^m c^m \lambda^{n-m} x^{2n+1-m\alpha}}{\Gamma(2n - m\alpha + 2)} \tag{39}$$

Comparing the graphs of the solutions numerically obtained by Formulas (37) and (39), we can draw a conclusion about their identity (Figure 3).

$$h(t, \alpha) := t - \sum_{n=1}^{50} \sum_{m=0}^{n} \left[ (-1)^{n+1} \frac{combin(n, m) c^m \lambda^{n-m} t^{2n+1-m\alpha}}{\Gamma(2n - m\alpha + 2)} \right]$$

### 3.1.3. Parametric Identification of the Model by the Experimental Data

The following technique for the parametric identification of parameter $\alpha$ is based on the experimental data, assuming that the rest of the parameters of the equation are known (with some degree of accuracy). This technique was developed due to the possibility of finding a solution at any point.

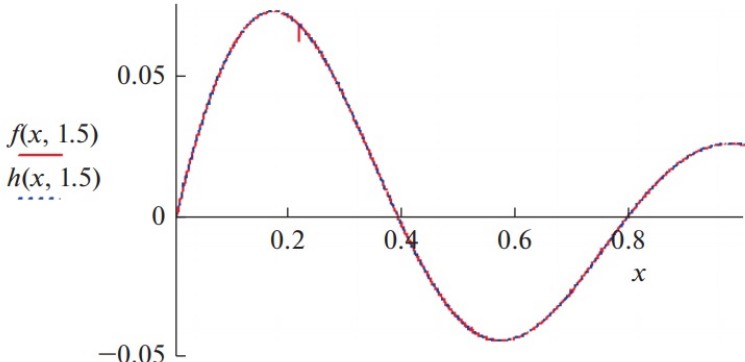

**Figure 3.** Determination of function by Formula (39) in Mathcad and comparison of this function with solution obtained using Laplace transform.

So, assume that we know several experimental points $u(x_i) = U_i$, $i = 1, ..., N$. To find the parameter $\alpha$ we minimize the deviation of the theoretical curves from the experimental curves. For calculating of the theoretical points we use Expression (33) for $u(x_i, \alpha)$. Using the least-squares method we determine the deviation function

$$F(\alpha) = \sum_{i=1}^{N} (U_i - u(x_i, \alpha))^2. \tag{40}$$

This function represents the sum of the deviations of the theoretical points from the experimental points. The value of $\alpha$ minimizing this function we can consider approximately as the search value. We shall note that the identification accuracy depends on the number of experimental points, together with the accuracy of the other parameters of the system. The method of parametric identification that we provide compares various nomographic techniques [39,40] (and references therein) and its advantage consists in the accurate quantitative estimation of choosing the search parameter. It is important that the deviation function (35) can be constructed on the entire range of supposed values of the parameter; this improves the accuracy of the identification. For testing the provided technique, we use the experimental data obtained in [35] (and references therein). The values for samples of polymer concrete based on polyester resin (dian- and dihloangidrid-1,1-dichloro-2,2-diethylene) are presented in Table 1.

**Table 1.** Experimental points for polymer concrete samples.

| $x_i(c)$ | 0.25 | 0.5 | 0.75 | 1 | 1.25 | 1.5 |
|---|---|---|---|---|---|---|
| $U_i$ | 0.05 | $-0.04$ | $-0.01$ | 0.02 | $-0.01$ | $-0.01$ |

Finally, using these data, we construct the deviation function (40) presented in Figure 4. The constructed graph shows that the deviation function has the minimum for $\alpha \approx 1.47$, and it allows us to assume that the order of the fractional derivative in Expression (31) is equal to 1.47. Figure 5 presents the experimental points and the theoretical curve. Comparing the experimental data presented in Table 1 with the model allows us to draw a conclusion that the provided model is adequate and our techniques for parametric identification have a high level of accuracy. The knowledge of the parameter $\alpha$ in the model (31) allows us, in particular, to predict various stress–strain characteristics of the material (polymer concrete, asphaltic concrete, etc.) when subjected to loading.

Now, having a technique for the parametric identification of the order of the fractional derivative in the Begley–Torvik model, we will proceed to the presentation of the method of separation of variables and its application to find the deformation-strength characteristics of polymer concrete.

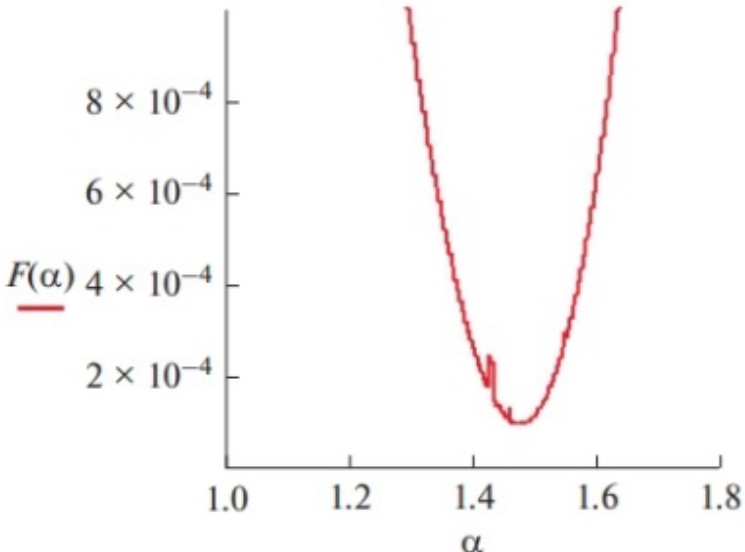

**Figure 4.** Deviation function (11) with parametric identification.

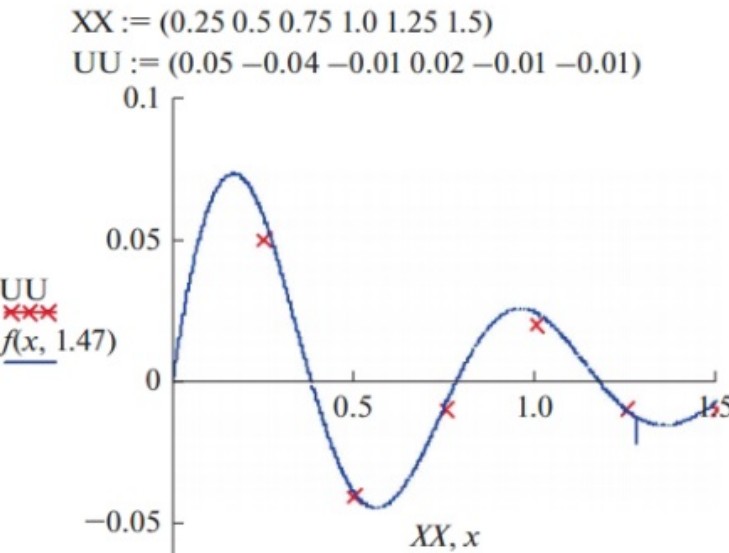

**Figure 5.** Comparison of experimental data with theoretical curve.

### 3.2. Method of Separation of Variables for Time–Space Fractional Vibration Equations

As it was noted in [41] the fractional calculus has attracted the attention of many authors in recent years. In this regard, we should note the paper [14], as a unique comprehensive review of fractional calculus and its application with the authoritative contribution of leading world experts. As it was noted earlier, we can describe anomalous diffusion or dispersion using the fractional space derivatives, and some processes with 'memory' effects using the fractional time derivatives. In [41] (and see the references therein), the following equation was investigated

$$\frac{\partial^2 u}{\partial t^2} = \frac{\partial^2 u}{\partial x^2} + C_0 D_{0t}^\alpha u + C_1 D_{0x}^\beta u + F$$

and was used to describe, in particular, the vibration of a string taking into account friction in a medium with fractal geometry. This equation may be used to model changes in the deformation-strength characteristics of polymer concrete under loading.

In domain $D = 0 < x < 1, 0 < t < 1$, we consider the first boundary value problem for the equation for the equation of vibration of a string with a fractional derivative of order with respect to the spatial variable

$$\frac{\partial^2 u}{\partial t^2} = \frac{\partial^2 u}{\partial x^2} + C_1 D_{0x}^\alpha u + C_0 D_{0t}^\beta u,\ 0 < \alpha,\ \beta < 2, \tag{41}$$

$$u(0,t) = u(1,t), \tag{42}$$

$$u(x,0) = \varphi(x), \tag{43}$$

$$u_t'(x,0) = \varphi(x), \tag{44}$$

Here, $0 < \alpha < 2$, $c$-constant, $D_{0x}^\alpha u$-constant, $D_{0x}^\alpha u$-fractional derivative of the Riemann–Liouville type of order $\alpha$. Fractional derivative of order $\alpha$ for function $f(x)$ in a point $x(0 \neq m - 1 < \alpha < m)$, $m \in N$) defined by the formula

$$D^\alpha f(x) = \frac{d^m}{dx^m}\left(\frac{1}{\Gamma(m-a)}\int_a^x \frac{f(\tau)}{(x-\tau)^{\alpha+1-m}}d\tau\right).$$

Obtained results are applied [14,41] for modeling changes in the deformation-strength characteristics of polymer concrete under loading. The solution to problem (41)–(44) will be sought by the Fourier method

$$u(x,t) = X(x)T(t). \tag{45}$$

We substitute (45) into Equation (41), then for an unknown function $X(x)$ we obtain the two-point Dirichlet problem

$$X''(x) + C_1 D_{0x}^\alpha X = \lambda X(x), \tag{46}$$

$$X(0) = X(1) = 0. \tag{47}$$

The solution to problem (46) and (47) was written out in [1,2], see the references therein. In particular, it was shown that the number $\lambda$ is an eigenvalue of problem (46) and (47), if and only if $\lambda$ is the zero of the function

$$\omega(\lambda) = \sum_{n=0}^{\infty}\sum_{k=0}^{n}\frac{C_n^k \lambda^{n-k}(-C_1)^k}{\Gamma(2n-k\beta+2)}$$

and the corresponding eigenfunctions $X_j(x)$ have the form

$$X_j(x) = \sum_{n=0}^{\infty}\sum_{k=0}^{n}\frac{C_n^k \lambda^{n-k}(-C_1)^k}{\Gamma(2n-k\beta+2)}x^{2n+1-k\alpha},\ j = 1,2,3,\ldots \tag{48}$$

(here $_j$-$j$-th eigenfunction of the problem (46) and (47)). The system of the eigenfunctions (48) is complete [14,41] but not orthogonal, thus we construct the system

$$\widetilde{X}_j(x) = (1-x) - \sum_{n=0}^{\infty}\sum_{k=0}^{n}\frac{C_n^k \lambda^{n-k}(-C_1)^k}{\Gamma(2n-k\beta+2)}x^{2n+1-k\alpha},\ j = 1,2,3,.. \tag{49}$$

which is biorthogonal to the system of eigenfunctions

$$X_j(x) = \sum_{n=0}^{\infty}\sum_{k=0}^{n}\frac{C_n^k \lambda^{n-k}(-C_1)^k}{\Gamma(2n-k\beta+2)}x^{2n+1-k\alpha},\ j = 1,2,3,\ldots$$

Next, we find the general solution to the equation

$$T(t) + c_0 D_{0t}^\beta T(t) = \lambda T(t).$$

As in the case of Equation (46), we have

$$T_m(t) = A_m \left( t + \sum_{n=0}^{\infty} \sum_{k=0}^{n} \frac{C_n^k \lambda^{n-k}(-C_1)^k}{\Gamma(2n - k\beta + 2)} x^{2n+1-k\alpha} \right) + B_m \left( 1 + \sum_{n=0}^{\infty} \sum_{k=0}^{n} \frac{C_n^k \lambda^{n-k}(-C_1)^k}{\Gamma(2n - k\beta + 1)} x^{2n+1-k\alpha} \right).$$

Let us designate

$$Z_m(t) = \left( t + \sum_{n=0}^{\infty} \sum_{k=0}^{n} \frac{C_n^k \lambda^{n-k}(-C_1)^k}{\Gamma(2n - k\beta + 2)} x^{2n+1-k\alpha} \right), \quad \widetilde{Z}_m(t) = \left( 1 + \sum_{n=0}^{\infty} \sum_{k=0}^{n} \frac{C_n^k \lambda^{n-k}(-C_1)^k}{\Gamma(2n - k\beta + 1)} x^{2n+1-k\alpha} \right).$$

Then the solution to problem (41)–(44) is written out in the standard way

$$u(x,t) = \sum_{m=1}^{\infty} T_m(t) X_m(x) = \sum_{m=1}^{\infty} [A_m Z_m(t) + B_m \widetilde{Z}_m(t)] X_m, \tag{50}$$

putting in the last expression $t - 0$, we have

$$\varphi(x) = \sum_{n=0}^{\infty} B_m \widetilde{Z}(0) Z(0) X_m(x),$$

so

$$B_m = \frac{1}{\widetilde{Z}(0)(\varphi(x), \widetilde{X}_m(x))(X_m(x), \widetilde{X}_m(x))}.$$

To find $A_m$ differentiate both parts (12) by $t$ and let $t = 0$ we obtain,

$$\sum_{m=1}^{\infty} [A_m Z'_m(0) + B_m \widetilde{Z}'_m(0)] X_m(x) = \psi(x)$$

from here

$$[A_m Z'_m(0) + B_m \widetilde{Z}'_m(0)](X_m(x), \widetilde{X}_m(x)) = (\psi(x), \widetilde{X}_m(x)).$$

Finally, we have,

$$A_m = \frac{1}{Z'_m(0)} \left[ \frac{1}{X_m(x), \widetilde{X}_m(x)} - B_m \widetilde{Z}'_m(0) \right]$$

which allows us to write a solution to problem (41)–(44) in the form (50).

In the earlier papers of the author, Equation (46) was used to model the certain deformation-strength characteristics of polymer concrete. In this paper, only transverse vibrations are considered, all movements occur in one plane and the granule moves perpendicular to the axis. Then, to simulate changes in the deformation-strength characteristics of polymer concrete under loading, we have the following first boundary-value problem (here $u(x,t)$-granule displacement in moment $t$)

$$\frac{\partial^2 u}{\partial t^2} = \frac{\partial^2 u}{\partial x^2} + C_0 D_{0x}^\beta u + C_1 D_{0t}^{1.47} u, \ 0 < \alpha, \ \beta < 2, \tag{51}$$

$$u(0,t) = u(1,t) = 0, \tag{52}$$

$$u(x,0) = \varphi(x) = 0, \tag{53}$$

$$u'_t(x,0) = \varphi(x) = 0, \tag{54}$$

the solution of which according to Formula (50) has the form

$$u(x,t) = \sum_{m=1}^{\infty} [A_m Z_m(t) + B_m \widetilde{Z}_m(t)] X_m,$$

where

$$X_j(x) = \sum_{n=0}^{\infty} \sum_{k=0}^{n} \frac{C_n^k \lambda^{n-k}(-C_1)^k}{\Gamma(2n - k\beta + 2)} x^{2n+1-1.47k},$$

$$\widetilde{X}_j(x) = (1-x) - \sum_{n=0}^{\infty} \sum_{k=0}^{n} \frac{C_n^k \lambda^{n-k}(-C_1)^k}{\Gamma(2n - k\beta + 2)} x^{2n+1-1.47k},$$

Let us find eigenvalues $\lambda_j$ numerically using the high-level language of technical calculations MATLAB taking $\alpha = 1.47$, $C_1 = 1.8$ (according by [41]). Eigenvalues are presented in Table 2.

**Table 2.** Numerical results for eigenvalues of the problem (41)–(44).

| $\lambda_1$ | $\lambda_2$ | $\lambda_3$ | $\lambda_4$ | $\lambda_5$ |
|---|---|---|---|---|
| 16.6 | 59.4 | 125.0 | 213.4 | 323.4 |

Then, an approximate solution to problem (51)–(54) will take the form

$$u(x,t) = \sum_{m=1}^{5} [A_m Z_m(t) + B_m \widetilde{Z}_m(t)] X_m, \tag{55}$$

Formula (55) allows us to write a solution to the problem (51)–(54) if the functions $\psi(x)$ and $\varphi(x)$ are continuously differentiable. Finally, it remains to determine the parameter $\beta$. This parameter can again be determined by the technique developed in [14,41] since the parameter $\alpha$ is already defined.

**Remark 3.** *We shall note the papers [42,43], where the same problem is solved by numerical methods and compared with the solution (55).*

**Funding:** This research received no external funding.

**Acknowledgments:** I am very thankful for reviewers for their very good grades and remarks, which made my manuscript better.

**Conflicts of Interest:** The author declares no conflict of interest. The funders had no role in the design of the study; in the collection, analyses, or interpretation of data; in the writing of the manuscript, or in the decision to publish the results.

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
