# Peer review of "Solving the Boundary Value Problems for Differential Equations with Fractional Derivatives by the Method of Separation of Variables"

_mathematics, doi:10.3390/math8111877_

Round 1
Reviewer 1 Report
The article addresses an interesting, topical topic, with a clear presentation and which is of interest to researchers in this field.
I consider that the approached topic could have been organized in two consecutive articles or with a little detail it could even be a book chapter in the presented field.
I recommend the article for publication in the journal Mathematics.
Author Response
Dear Reviewer!
Of course, I agree that this manuscript with your recommendation about two consecutive articles or with a little detail it could even be a book chapter in the presented field. But (unfortunately) I'm afraid I can't finish it..
Huge thank you for your recomendations.
Reviewer 2 Report
See the enclosed Report

Author Response
Dear Reviewer!
All your remarks I took into account and in the text of the manuscript overwritten corrections with blue color. I'd like to talk about remark
" page 11, lines 150-151
The string
... (where ρ > 1=2, ρ ≡ 1, Im(µ) = 0)
seems to be not correctly written, with respect to ρ "
This mistake I corrected too.
Dear Reviewer, thank you very much for such high grade for my manuscript. It's very and very important for me.
Reviewer 3 Report
Dear authors,
Please consider the suggested comments to improve the quality of the current version of the manuscript:
- In the abstract please mention the scope of the paper including the limitations of the current/previous studies performed by the peers in this field and specify the advances made to overcome those limitations.
- In the abstract, the list of contributions was listed. However, the necessary logical reasoning to support the statements listed in contributions was not mentioned (line 4-6). Please include the details and scientific reasons for the statements listed in contributions.
- In the abstract, kindly incorporate the comparison study of results to have the reader understand the details of the results and the magnitude of the results obtained. For example, in line 5-6, it was mentioned that the information provided by the spectral theory is undoubtedly of independent interest. But the parameters considered for comparison to obtain the conclusions were not described in detail. Please include the necessary logical reasons considered to obtain the conclusions made.
- Very importantly the abstract of a scientific research paper should be precisely mentioning the specific research question that is answered, conditions used to obtain the data sets, parameters, results, and conclusions. Please update the abstract with the parameters considered, results obtained, and scientific conclusions driven, as suggested.
- In the introduction, please include the knowledge gaps existing in the current research work and prior studies performed in the field, with appropriate references. Please specify and establish the need for the current work presented in the manuscript.
- In the last paragraph of the introduction, kindly include the details of the broader impacts on the study made and the results achieved. It is very important to provide the future scope of the research performed to make a strong impact on the readers on the research performed.
- In section 2: Method of separation of variables for Time-Space Fractional Vibration Equations - the basic theory: Please include the logical explanation of the outcome of results needs to be incorporated, especially figure 2, 3, 4, and 5. Also, please include the necessary arguments with the ongoing research results from the peers with appropriate references.
- Please use the proper format for writing and mentioning the mathematical equations. Kindly refer to the author's instructions to understand the formatting process during the submission of the manuscript. Please use the link below for author instructions: https://www.mdpi.com/journal/mathematics/instructions
- Please revise the manuscript with English grammar to improve the quality of the manuscript with respect to English writing.
Author Response
Dear Reviewer!
Thank you very much. I rewritten the abstract according to your suggestions. In the Introduction (where you asked to note contribution of other researchers) I provided the refer the paper of M.Dzhrbashian and on page 13 - papers of E.Larionov. There investigated problems of the completteness of eigenfunctions and associaated functions.
Once again I'd like to note, that spectral sctucture of operators in this manuscript is very poorly investigated (the papers of Dzhrbashian, Larionov, Malamud I refer).
There was very important remark about graphs. According to your suggestions, I put them in correct order.
For preparing the manuscript, accordint to the Author's Rules I use the Overleaf system ( for Latex) with the template for Mathematics ( https://www.mdpi.com/authors/latex)
Once again, dear Reviewer, many thanks you for your review.
Round 2
Reviewer 3 Report
Dear authors,
Thank you for updating the manuscript with recommended changes.